# Type 2 diabetes and its genetic susceptibility are associated with increased severity and mortality of COVID-19 in UK Biobank

Aeyeon Lee[1], Jieun Seo[1], Seunghwan Park[1,2], Youngkwang Cho[1], Gaeun Kim[1], Jun Li [3], Liming Liang [4,5,7✉], Taesung Park [6,7✉] & Wonil Chung [1,5,7✉]

Type 2 diabetes (T2D) is known as one of the important risk factors for the severity and mortality of COVID-19. Here, we evaluate the impact of T2D and its genetic susceptibility on the severity and mortality of COVID-19, using 459,119 individuals in UK Biobank. Utilizing the polygenic risk scores (PRS) for T2D, we identified a significant association between T2D or T2D PRS, and COVID-19 severity. We further discovered the efficacy of vaccination and the pivotal role of T2D-related genetics in the pathogenesis of severe COVID-19. Moreover, we found that individuals with T2D or those in the high T2D PRS group had a significantly increased mortality rate. We also observed that the mortality rate for SARS-CoV-2-infected patients was approximately 2 to 7 times higher than for those not infected, depending on the time of infection. These findings emphasize the potential of T2D PRS in estimating the severity and mortality of COVID-19.

[1] Department of Statistics and Actuarial Science, Soongsil University, Seoul 06978, Korea. [2] Institute of Genetic Epidemiology, Basgenbio Co. Ltd., Seoul 04167, Korea. [3] Department of Nutrition, Harvard T.H. Chan School of Public Health, Boston, MA 02115, USA. [4] Department of Biostatistics, Harvard T.H. Chan School of Public Health, Boston, MA 02115, USA. [5] Program in Genetic Epidemiology and Statistical Genetics, Harvard T.H. Chan School of Public Health, Boston, MA 02115, USA. [6] Department of Statistics, Seoul National University, Seoul 08826, Korea. [7] These authors jointly supervised this work: Liming Liang, Taesung Park, Wonil Chung. ✉email: lliang@hsph.harvard.edu; tspark@stats.snu.ac.kr; wchung@ssu.ac.kr

The coronavirus disease 2019 (COVID-19) is caused by an infection with the severe acute respiratory syndrome coronavirus 2 (SARS-CoV-2) and remains a global pandemic[1]. COVID-19 affects the respiratory system and other crucial bodily systems, presenting a spectrum of symptoms ranging from asymptomatic and mild cases to severe illnesses, including severe pneumonia, acute respiratory distress syndrome (ARDS), and even death[2–4]. Furthermore, a considerable portion of COVID-19 patients, at least one-third, develop what is termed post-acute COVID-19 syndrome, which encompasses respiratory complications, fatigue, nervous system disturbances, and musculoskeletal issues[5]. These persist for ~4 weeks or more following the initial onset of symptoms[6]. Despite rigorous vaccination efforts, the potential for severe and chronic outcomes of COVID-19 remains. This underscores the urgent need for enhanced characterization of individuals at a heightened risk for clinically serious disease outcomes.

A variety of studies have indicated that certain clinical risk factors are associated with increased severity of COVID-19, often leading to hospitalization or death. These factors include older age, male gender, smoking, and lower socioeconomic status[7,8]. Prevalent comorbidities, such as obesity[9–12], type 2 diabetes (T2D)[13,14], respiratory diseases, hypertension, cardiovascular disease (CVD)[15,16], and dementia[17], also correlate with poorer COVID-19 outcomes. Moreover, an individual's genetic makeup significantly influences the degree of COVID-19 severity[18]. This is supported by extensive population-based genome-wide association studies (GWAS), which suggest that a number of common genetic variants can affect the severity of symptoms following a COVID-19 infection[4,19]. The genes most strongly associated with severe COVID-19 progression, as identified by GWAS, typically play roles in immune response mechanisms, lung disease pathology, CVD[20], and T2D[21].

Each individual variant associated with COVID-19 severity, identified in the original GWAS, has a minor effect when considered separately, but cumulatively, they contribute significantly to the genetic variation seen in the severity of COVID-19. By integrating these variants, such as single nucleotide polymorphisms (SNPs), into a polygenic risk score (PRS), we can more effectively identify patients at potential risk within a cohort. The COVID-19 PRS has been known to be associated with an increased risk of severity and morbidity from COVID-19 across multiple independent groups[22,23]. PRSs for other underlying diseases, such as obesity, asthma, and schizophrenia, have also been found to correlate with COVID-19 severity or disease progression[23–25]. However, most studies have primarily focused on the association between COVID-19 PRS or PRSs for other underlying diseases and the severity or clinical progression of the illness. Research on the effect of PRS on COVID-19 mortality is limited. Notably, only a few studies have assessed the impact of T2D or T2D PRS on the severity and mortality of COVID-19, despite T2D being recognized as one of the comorbidities strongly associated with severe COVID-19.

In this paper, we systematically investigate the association of T2D and T2D PRS with the severity and mortality of COVID-19. We also evaluate the effects of various SARS-CoV-2 variants, including the recent Omicron variants, and assess the impact of vaccinations. First, we estimate the genetic predisposition for T2D by computing the T2D PRS using GWAS summary statistics from the UK Biobank. We include body mass index (BMI) as a covariate in the GWAS when constructing the T2D PRS, to mitigate the confounding effect of BMI on T2D. This approach is informed by the Mendelian randomization (MR) studies on the causal effects of T2D on COVID-19 severity, which have yielded inconsistent results, largely attributable to the confounding impact of BMI on T2D[26,27]. Next, we assess the relationship between T2D, T2D PRS, and the severity of COVID-19 using proportional odds models. We then investigate the effect on survival times of T2D patients infected with SARS-CoV-2 using Cox proportional hazards models with time-dependent coefficients[28] for SARS-CoV-2 infection to satisfy the proportional hazards assumption. In this model, we incorporate either T2D or T2D PRS to determine the impact of T2D or genetic risks for T2D on survival time. Additionally, we examine differences in survival times across SARS-CoV-2 variants and vaccination statuses. Lastly, we conduct a stratified survival analysis for COVID-19, T2D, and T2D PRS groups, and perform the pairwise log-rank test to identify significant differences in mortality between these groups.

## Results

**Ascertainment of T2D and SARS-CoV-2 cases**. The overall study design is shown in Fig. 1 and the baseline characteristics for T2D and T2D-related traits are detailed in Table 1. Based on the recently updated datasets from the UK Biobank (up to December 2022), we ascertained the number of T2D cases and SARS-CoV-2 confirmed cases from 459,119 participants. We identified a total of 37,110 (8.1%) T2D patients from the self-reported data (data field 2443), hospital inpatient records, and death data (International Classification of Diseases, 10th Revision (ICD-10), code E11) (Fig. 2a). We obtained a total of 101,271 (22.1%) SARS-CoV-2 confirmed cases from the COVID-19 test data, hospital inpatient records, and death data (ICD-10, codes U07.1 and U07.4) (Fig. 2b).

We prepared two COVID-19-related phenotypes: (1) all cases with reported SARS-CoV-2 infection regardless of symptoms (101,271 cases and 357,848 controls) and (2) individuals with severe COVID-19, defined as those who were hospitalized or died due to symptoms associated with the infection (7478 cases and 451,641 controls). Based on the SARS-CoV-2 infection dates and information on the proportions of all variants during the study period in the UK, we classified participants into seven variant groups: early variants, EU1 (20E), Alpha (20I), Delta (21J), Omicron1 (21K), Omicron2 (21L), and Omicron3 (22B) variants, defining early variants as all variants detected in the UK before the EU1 variant appeared. We randomly assigned all SARS-CoV-2-infected cases to one of the variant groups according to the proportions of variants on their SARS-CoV-2 infection date. The vaccination data contains self-reported vaccination status (data fields 27,983, 27,985) and dates (data fields 27,984, 27,986). The unvaccinated group ($N = 279,223$) was defined as participants who had no vaccination record, as well as those diagnosed with COVID-19 prior to vaccination or within 14 days of receiving the first dose[29].

**Heritability and genetic correlation**. We performed GWAS analyses for T2D, SARS-CoV-2 infection, and severe COVID-19 (Supplementary Fig. 1), and then estimated heritability and genetic correlations between various phenotypes based on GWAS summary statistics. Supplementary Table 1 shows the estimated heritability in the diagonal elements and the estimated genetic correlation between pairs of traits in non-diagonal elements. The estimated heritability for T2D and severe COVID-19 were 0.154 (SE = 0.012) and 0.027 (SE = 0.009), respectively, implying that the phenotypic variation of T2D and severe COVID-19 can be explained approximately 15.4% and 2.7% by SNPs, respectively. The estimated genetic correlation between T2D and severe COVID-19 was 0.242 (SE = 0.086), indicating a potential shared genetic architecture between T2D and severe COVID-19. The estimated heritability and genetic correlation for other traits are described in Supplementary Note 1.

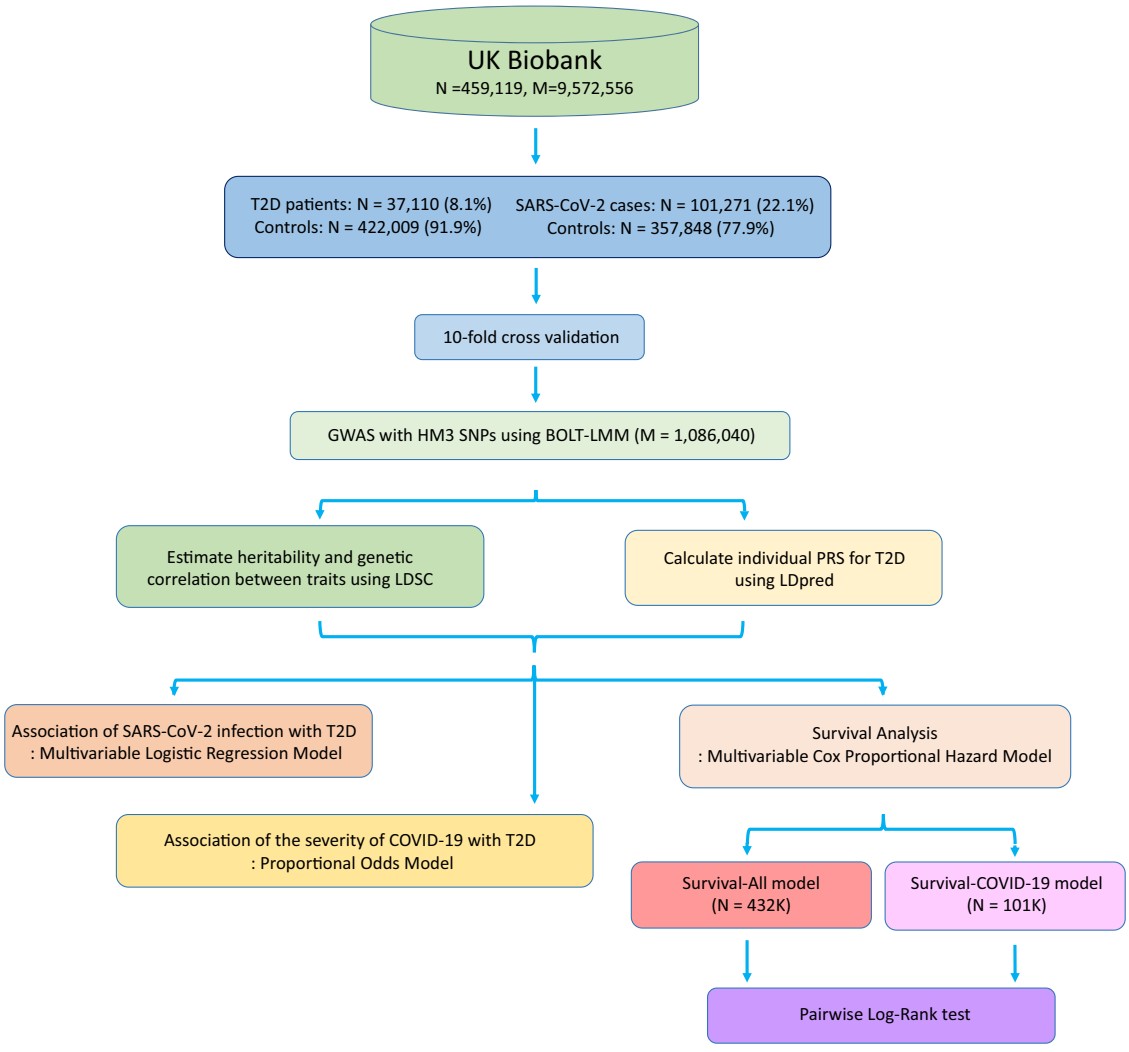

**Fig. 1 The overview of study.** We utilized 459,119 individuals in the UK Biobank to evaluate the impact of T2D and its genetic susceptibility on the severity and mortality of COVID-19. We calculated polygenic risk scores (PRS) based on summary statistics from the UK Biobank to capture the overall genetic susceptibility to T2D using 10-fold cross-validation. In our investigation, we examined the association of T2D and T2D PRS with COVID-19 severity using proportional odds models. We further explored the effect of T2D and T2D PRS on survival times in COVID-19 patients using Cox proportional hazard models.

**Genetic susceptibility**. For the 459,119 participants, we computed individual PRS representing the genetic susceptibility for traits using a 10-fold cross-validation (CV), and then standardized the PRS. We further derived scores of adjusted PRS based on logistic or linear regression models with PRS, age, and gender as covariates. The estimated area under the curves (AUCs) of T2D PRS and severe COVID-19 were 0.641 and 0.518, and those of the adjusted T2D PRS and COVID-19 were 0.711 and 0.644, respectively (Supplementary Table 2). Supplementary Fig. 2 displays the ROC curves for PRS and adjusted T2D PRS and Supplementary Table 3 shows the estimated AUC for the candidate PRS across a range of tuning parameters. The PRS with the best discriminative capability is highlighted in bold type.

**Association between the severity of COVID-19 and T2D**. In order to identify the association between the severity of COVID-19 and T2D or its genetic susceptibility, we divided SARS-CoV-2-infected participants into three groups according to the severity of their COVID-19 symptoms: mild (1), moderate (2), and critical (3). We checked the suitability of the model by testing the proportional odds assumption based on the Brant test and found that

the proportional odds model fit our data (Table 2). The associations of T2D and its genetic susceptibility with the severity of COVID-19 using the proportional odds model are described in Table 2. The estimated odds ratio (OR) of T2D was 2.824 (95% CI = 2.644–3.016), implying that the odds of COVID-19 severity among T2D patients was 2.824 times higher than in controls.

We included T2D PRS as a covariate and the estimated OR of T2D PRS was 1.080 (95% CI = 1.050–1.111), meaning that the odds of COVID-19 severity among individuals with a one-unit increase (=standard deviation [SD]) in T2D PRS were 1.080 times greater than those of the controls. When we included the PRS group as a covariate (with low PRS group as the reference), the estimated OR of the high PRS group compared to the low PRS group was 1.211 (95% CI = 1.102–1.330), meaning that the odds of COVID-19 severity in the high PRS group were 1.211 times greater than that in the low PRS group. We also included vaccination status as a covariate and found that the estimated OR of vaccination status was 0.206 (95% CI = 0.193–0.220), indicating that the odds of COVID-19 severity in the vaccinated group was 0.206 times less than that in the non-vaccinated group.

**Table 1 Baseline characteristics for T2D, BMI and WHR in 459,119 participants in UK Biobank.**

| Covariates | Group | N | T2D | | BMI | | WHR | | NA |
|---|---|---|---|---|---|---|---|---|---|
| | | | #Case | Prev | Mean | SD | Mean | SD | |
| Age | | 459,119 | 37,110 | 0.081 | 27.40 | 4.77 | 0.87 | 0.09 | 0 |
| | <60 | 156,754 | 6909 | 0.044 | 27.09 | 4.97 | 0.86 | 0.09 | |
| | ≥60 | 302,365 | 30,201 | 0.100 | 27.55 | 4.66 | 0.88 | 0.09 | |
| | p-value | | $<2 \times 10^{-16}$ | | $<2 \times 10^{-16}$ | | $<2 \times 10^{-16}$ | | |
| Sex | | 459,119 | 37,110 | 0.081 | 27.40 | 4.77 | 0.87 | 0.09 | 0 |
| | Female | 249,227 | 14,659 | 0.059 | 27.01 | 5.14 | 0.82 | 0.07 | |
| | Male | 209,892 | 22,451 | 0.107 | 27.85 | 4.24 | 0.94 | 0.07 | |
| | p-value | | $<2 \times 10^{-16}$ | | $<2 \times 10^{-16}$ | | $<2 \times 10^{-16}$ | | |
| BMI | | 457,620 | 36,842 | 0.081 | 27.40 | 4.77 | 0.87 | 0.09 | 1499 |
| | <25 | 152,107 | 3273 | 0.022 | 22.78 | 1.65 | 0.82 | 0.07 | |
| | ≥25& < 30 | 194,843 | 12,487 | 0.064 | 27.29 | 1.40 | 0.88 | 0.08 | |
| | ≥30 | 110,670 | 21,082 | 0.190 | 33.92 | 3.86 | 0.92 | 0.09 | |
| | p-value | | $<2 \times 10^{-16}$ | | $<2 \times 10^{-16}$ | | $<2 \times 10^{-16}$ | | |
| Smoking status | | 457,606 | 36,910 | 0.081 | 27.39 | 4.77 | 0.87 | 0.09 | 1513 |
| | Never | 248,025 | 15,530 | 0.063 | 27.12 | 4.75 | 0.86 | 0.09 | |
| | Previous | 163,610 | 16,629 | 0.102 | 27.90 | 4.73 | 0.89 | 0.09 | |
| | Current | 45,971 | 4751 | 0.103 | 27.05 | 4.81 | 0.89 | 0.09 | |
| | p-value | | $<2 \times 10^{-16}$ | | $<2 \times 10^{-16}$ | | $<2 \times 10^{-16}$ | | |
| Alcohol intake | | 458,806 | 37,069 | 0.081 | 27.40 | 4.77 | 0.87 | 0.09 | 313 |
| | Never | 32,335 | 4668 | 0.144 | 28.18 | 5.72 | 0.87 | 0.09 | |
| | 1–3 times/month | 102,911 | 11,260 | 0.109 | 28.26 | 5.47 | 0.86 | 0.09 | |
| | 1 or 2 times/week | 121,099 | 9095 | 0.075 | 27.46 | 4.65 | 0.87 | 0.09 | |
| | 3 or 4 times/w | 109,316 | 6302 | 0.058 | 26.90 | 4.21 | 0.87 | 0.09 | |
| | Daily | 93,145 | 5744 | 0.062 | 26.69 | 4.11 | 0.88 | 0.09 | |
| | p-value | | $<2 \times 10^{-16}$ | | $<2 \times 10^{-16}$ | | $<2 \times 10^{-16}$ | | |
| Townsend index | | 458,578 | 37,061 | 0.081 | 27.40 | 4.77 | 0.87 | 0.09 | 541 |
| | <Q1 | 114,510 | 7180 | 0.063 | 26.96 | 4.33 | 0.86 | 0.09 | |
| | ≥Q1 & <Q2 | 114,784 | 8019 | 0.070 | 27.21 | 4.51 | 0.87 | 0.09 | |
| | ≥Q2 & <Q3 | 114,644 | 9083 | 0.079 | 27.42 | 4.76 | 0.87 | 0.09 | |
| | ≥Q3 | 114,640 | 12,779 | 0.111 | 28.00 | 5.36 | 0.88 | 0.09 | |
| | p-value | | $<2 \times 10^{-16}$ | | $<2 \times 10^{-16}$ | | $<2 \times 10^{-16}$ | | |

Townsend index: area deprivation.
*Prev* prevalence, *SD* standard deviation, *WHR* Waist-Hip Ratio.

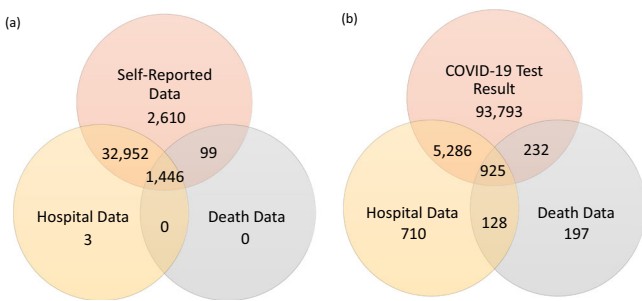

**Fig. 2 The sample sizes for T2D cases and SARS-CoV-2 confirmed cases across three datasets such as self-reported data, hospital inpatient data and death data. a** The total sample size for T2D patients is 37,110, identified from self-reported data, hospital inpatient data, and death data. **b** The total sample size for SARS-CoV-2 confirmed cases is 101,271, derived from laboratory-confirmed PCR test results, hospital inpatient data, and death data.

**Survival analysis with all participants**. To investigate the effect of T2D on survival times in patients with SARS-CoV-2 infection, we conducted survival analysis with all participants (Survival-All). We performed a multivariable Cox proportional hazards (PH) model including T2D and time-dependent coefficients for SARS-CoV-2 infection adjusted for other clinical variables, after confirming that the proportional hazards assumption was not violated for any covariates (Supplementary Table 4, Supplementary

Figs. 3 and 4). The hazard ratio (HR) for T2D was 2.266 (95% CI = 2.164–2.372), implying that the risk of death for T2D patients was 2.266 times higher than for controls at any point in time, holding all other covariates constant (Table 3). The HR for SARS-CoV-2 infection within five intervals (i.e., before Delta, Delta, Omicron 1, 2, 3) were 7.333, 2.013, 1.864, 1.906, and 1.986, respectively. This means that the risks of death for SARS-CoV-2-infected individuals were 7.333 (95% CI = 6.902–7.791), 2.013 (95% CI = 1.811–2.238), 1.864 (95% CI = 1.636–2.123), 1.906 (95% CI = 1.692–2.146), and 1.986 (95% CI = 1.744–2.262) times higher than that for controls within five intervals, respectively (Table 3). The variable SARS-CoV-2 infection significantly affected all five intervals, with the effect in the first interval being much higher than that in the other intervals as expected. Figure 3a displays the estimated survival curves stratified with T2D and SARS-CoV-2 infection based on the Survival-All model. Four groups (i.e., controls, T2D patients, COVID-19 patients, T2D patients with SARS-CoV-2 infection) had significantly different ($p < 0.05$) survival probabilities based on pairwise log-rank tests. In general, SARS-CoV-2 infection led to decreased survival probability compared to controls, and SARS-CoV-2-infected T2D patients had dramatically decreased survival probabilities compared to other groups.

We next included adjusted T2D PRS as a covariate in the Cox PH model, with adjustments for the clinical variables, to examine the effect of high genetic risk for T2D and SARS-CoV-2 infection on survival times. The HR of T2D PRS was 1.062 (95% CI = 1.042–1.081), indicating that the risk of death for individuals

**Table 2 The results of proportional odds models for the association of severity of COVID-19 and T2D.**

| Covariates | OR (95% CI) | p-value | Brant test |
|---|---|---|---|
| T2D + Clinical | T2D: 2.824 (2.644–3.016) | $2.95 \times 10^{-209}$ | 0.51 |
| T2D PRS + Clinical | T2D PRS: 1.080 (1.050–1.111) | $8.58 \times 10^{-8}$ | 0.24 |
| T2D PRS Group + Clinical | Medium: 1.083 (1.015–1.156) | $1.66 \times 10^{-2}$ | 0.82 |
| | High: 1.211 (1.102–1.33) | $7.01 \times 10^{-5}$ | 0.44 |
| T2D + Vaccine + Clinical | T2D: 2.517 (2.353–2.693) | $1.69 \times 10^{-158}$ | 0.13 |
| | Vaccine: 0.206 (0.193–0.220) | $2.23 \times 10^{-308}$ | 0.06 |

Clinical: age, gender, BMI, genotyping array and PC 1–4. T2D PRS: genetic susceptibility for T2D adjusted for age and gender using the logistic regression model. T2D PRS Group: low, medium and high PRS groups for T2D. $H_0$ of Brant test: model holds on the assumption of constant effect size of a risk factor.

**Table 3 The results of Cox proportional hazards model with all participants to identify the effects of T2D and SARS-CoV-2 infection on mortality for 432K participants from UK Biobank.**

| Covariates | HR (95% CI) | p-value |
|---|---|---|
| T2D + Clinical | T2D: 2.393 (2.286–2.505) | $4.82 \times 10^{-307}$ |
| SARS-CoV-2 + Clinical | time-group1: 7.763 (7.311–8.244) | $2.23 \times 10^{-308}$ |
| | time-group2: 2.057 (1.850–2.286) | $1.22 \times 10^{-40}$ |
| | time-group3: 1.883 (1.653–2.145) | $1.82 \times 10^{-21}$ |
| | time-group4: 1.924 (1.708–2.166) | $3.41 \times 10^{-27}$ |
| | time-group5: 2.006 (1.761–2.285) | $9.87 \times 10^{-26}$ |
| T2D + SARS-CoV-2 + Clinical | T2D: 2.266 (2.164–2.372) | $3.45 \times 10^{-267}$ |
| | time-group1: 7.333 (6.902–.791) | $2.23 \times 10^{-308}$ |
| | time-group2: 2.013 (1.811–2.238) | $2.61 \times 10^{-38}$ |
| | time-group3: 1.864 (1.636–2.123) | $7.54 \times 10^{-21}$ |
| | time-group4: 1.906 (1.692–2.146) | $1.82 \times 10^{-26}$ |
| | time-group5: 1.986 (1.744–2.262) | $4.83 \times 10^{-25}$ |
| T2D PRS + Clinical | T2D PRS: 1.067 (1.047–1.087) | $4.95 \times 10^{-12}$ |
| T2D PRS + SARS-CoV-2 + Clinical | T2D PRS: 1.062 (1.042–1.081) | $1.31 \times 10^{-10}$ |
| | time-group1: 7.744 (7.292–8.223) | $2.23 \times 10^{-308}$ |
| | time-group2: 2.055 (1.849–2.285) | $1.42 \times 10^{-40}$ |
| | time-group3: 1.882 (1.652–2.144) | $1.90 \times 10^{-21}$ |
| | time-group4: 1.923 (1.708–2.166) | $3.50 \times 10^{-27}$ |
| | time-group5: 2.006 (1.761–2.285) | $9.94 \times 10^{-26}$ |
| T2D PRS Group + Clinical | Medium: 1.077 (1.025–1.131) | $3.19 \times 10^{-3}$ |
| | High: 1.192 (1.116–1.273) | $1.53 \times 10^{-7}$ |
| T2D PRS Group + SARS-CoV-2 + Clinical | Medium: 1.075 (1.024–1.130) | $3.74 \times 10^{-3}$ |
| | High: 1.181 (1.105–1.261) | $7.37 \times 10^{-7}$ |
| | time-group1: 7.753 (7.301–8.233) | $2.23 \times 10^{-308}$ |
| | time-group2: 2.056 (1.849–2.285) | $1.39 \times 10^{-40}$ |
| | time-group3: 1.882 (1.652–2.144) | $1.89 \times 10^{-21}$ |
| | time-group4: 1.923 (1.708–2.166) | $3.48 \times 10^{-27}$ |
| | time-group5: 2.006 (1.761–2.285) | $9.92 \times 10^{-26}$ |
| T2D + SARS-CoV-2 + Vaccine + Clinical | T2D: 2.093 (2.000–2.191) | $4.39 \times 10^{-220}$ |
| | Vaccine: 0.198 (0.186–0.210) | $2.23 \times 10^{-308}$ |
| | time-group1: 5.348 (5.031–5.684) | $2.23 \times 10^{-308}$ |
| | time-group2: 1.799 (1.618–2.001) | $1.90 \times 10^{-27}$ |
| | time-group3: 1.871 (1.642–2.131) | $4.66 \times 10^{-21}$ |
| | time-group4: 1.986 (1.764–2.237) | $1.06 \times 10^{-29}$ |
| | time-group5: 2.097 (1.841–2.389) | $7.15 \times 10^{-29}$ |

Clinical: age, gender, BMI, genotyping array, PC 1–4. The variable time-group identifies which time-interval each row belongs to. That is, time-group = 1 identifies the first time-interval (0, 494], time-group = 2 the second time-interval (494, 672], time-group = 3 the third time-interval (672, 757], time-group = 4 the fourth time-interval (757, 851] and time-group = 5 the fifth time-interval (851, 1014].
HR hazard ratio, CI confidence interval.

with a one-unit increase (=SD) in T2D PRS is 1.062 times higher than for controls (Table 3). This suggests that genetic susceptibility for T2D increases the risk of death. We further examined the effect of PRS group for T2D and SARS-CoV-2 infection on survival times. As in the proportional odds models, we included the PRS group as a covariate in the model, adjusting for potential confounders. The HR for the high PRS group was 1.181 (95% CI = 1.105–1.261), indicating that the risk of death for individuals in the high PRS group for T2D was 1.181 times greater than that

for individuals in the low PRS group for T2D (Table 3). Figure 3b displays the estimated survival curves stratified with PRS group and SARS-CoV-2 infection, based on the Survival-All model. The survival probabilities for individuals without SARS-CoV-2 infection were similar among the three PRS groups, but the three PRS groups with SARS-CoV-2 infection had significantly distinct survival probabilities based on pairwise log-rank tests. SARS-CoV-2 infection generally decreased survival probabilities and, particularly, COVID-19 patients in high PRS group had the

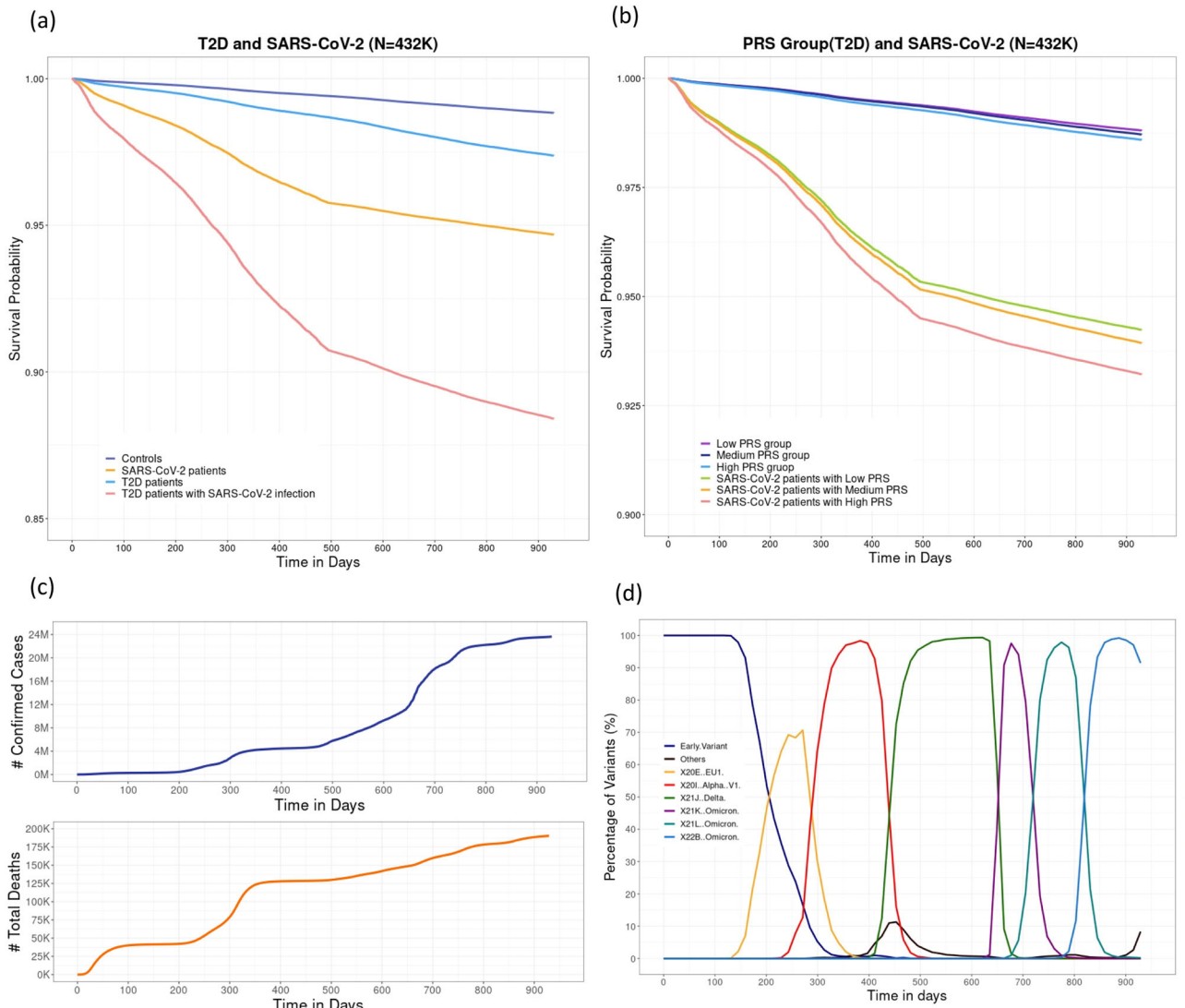

**Fig. 3 The survival curves stratified with T2D or T2D PRS and SARS-CoV-2 infection, based on the Survival-All model and the number of SARS-CoV-2 confirmed cases, total deaths and the proportion of SARS-CoV-2 variants in UK. a** The survival curves stratified by T2D and SARS-CoV-2 infection (N = 432K) based on the Survival-All model, adjusted for clinical covariates such as age, BMI, gender, genotyping array, and PC 1–4. **b** The survival curves stratified by PRS groups for T2D and SARS-CoV-2 infection (N = 432K) based on the Survival-All model, adjusted for clinical covariates. **c** The number of SARS-CoV-2 confirmed cases and total deaths due to COVID-19. **d** The proportion of SARS-CoV-2 variants in England during the study period.

lowest survival probability, followed by the medium and low PRS groups. This indicates the potential genetic role of T2D in SARS-CoV-2 infection.

Figure 3c shows the number of confirmed COVID-19 cases and total deaths due to COVID-19, while Fig. 3d displays the proportion of COVID-19 variants in the UK during the study period. Survival probabilities for all groups considerably decreased between 230 and 470 days in Fig. 3a and b. This can be explained by the rapid increase in confirmed COVID-19 cases and deaths in the UK during the surge of the COVID-19 Alpha variant. The dominant COVID-19 variants transitioned to Alpha (red curve), Delta (green), and Omicron variants (purple, turquoise, and blue) as shown in Supplementary Fig. 5. The Alpha variants were more difficult to spread compared to the Delta variants, but the fatality rate of Alpha variants was generally higher than Delta variants, leading to a considerable increase in deaths during the Alpha variant surge in the UK. As in the proportional odds models, we included vaccination status as a covariate. The estimated HR of vaccination status was 0.198 (95% CI = 0.186–0.210), meaning that the risk of death for individuals

in the vaccinated group was 0.198 times less than that in the non-vaccinated group (Table 3).

**Survival analysis with SARS-CoV-2-infected Individuals**. We restricted our study to only SARS-CoV-2-infected individuals (N = 101,086) to evaluate the effect of T2D on survival times in COVID-19 positive patients (Survival-COVID-19). We performed a multivariable Cox PH model similar to the Survival-All model after confirming that the proportional hazards assumption was not violated for all covariates (Supplementary Figs. 6 and 7). We found similar results for the HR of T2D, T2D PRS, PRS groups, and vaccination status to the Survival-All model (Table 4). Instead of including time-dependent coefficients for SARS-CoV-2 infection, we included an indicator for the variants (i.e., early, EU1, Alpha, Delta, Omicron1,2,3) as covariates in the model. Compared to the Omicron3 variant (reference), the HRs of Early, EU1, Alpha, Delta, Omicron1,2 variants were 7.993 (95% CI = 6.636–9.628), 4.454 (95% CI = 3.665–5.414), 5.291 (95% CI = 4.358–6.424), 1.359 (95% CI = 1.117–1.654), 0.847 (95%

**Table 4 The results of Cox proportional hazards model with SARS-CoV-2-infected individuals to identify the effect of T2D on survival times only for 101K SARS-CoV-2 confirmed cases.**

| Covariates | HR (95% CI) | P-value |
|---|---|---|
| T2D + Clinical | T2D: 2.360 (2.154–2.584) | $2.67 \times 10^{-76}$ |
| T2D PRS + Clinical | T2D PRS: 1.062 (1.022–1.104) | $1.99 \times 10^{-3}$ |
| T2D PRS Group + Clinical | Medium: 1.089 (0.977–1.214) | $1.23 \times 10^{-1}$ |
| | High: 1.190 (1.034–1.370) | $1.55 \times 10^{-2}$ |
| T2D + SARS-CoV-2 variants + Clinical | T2D: 1.973 (1.801–2.161) | $1.75 \times 10^{-48}$ |
| | Early: 7.993 (6.636–9.628) | $2.93 \times 10^{-106}$ |
| | EU1: 4.454 (3.665–5.414) | $6.72 \times 10^{-51}$ |
| | Alpha: 5.291 (4.358–6.424) | $1.61 \times 10^{-63}$ |
| | Delta: 1.359 (1.117–1.654) | $2.17 \times 10^{-3}$ |
| | Omicron1: 0.847 (0.689–1.040) | $1.13 \times 10^{-1}$ |
| | Omicron2: 0.789 (0.644– 0.968) | $2.29 \times 10^{-2}$ |
| T2D PRS + SARS-CoV-2 variants + Clinical | T2D PRS: 1.045 (1.006– 1.086) | $2.31 \times 10^{-2}$ |
| | Early: 8.316 (6.905–10.016) | $1.97 \times 10^{-110}$ |
| | EU1: 4.590 (3.776–5.578) | $6.26 \times 10^{-53}$ |
| | Alpha: 5.483 (4.516–6.657) | $3.14 \times 10^{-66}$ |
| | Delta: 1.350 (1.110–1.643) | $2.70 \times 10^{-3}$ |
| | Omicron1: 0.837 (0.681–1.028) | $8.97 \times 10^{-2}$ |
| | Omicron2: 0.781 (0.637– 0.957) | $1.72 \times 10^{-2}$ |
| T2D PRS Group + SARS-CoV-2 variants + Clinical | Medium: 1.093 (0.98–1.219) | $1.09 \times 10^{-1}$ |
| | High: 1.145 (0.994–1.319) | $6.12 \times 10^{-2}$ |
| | Early: 8.328 (6.915–10.03) | $1.40 \times 10^{-110}$ |
| | EU1: 4.596 (3.781–5.585) | $5.06 \times 10^{-53}$ |
| | Alpha: 5.491 (4.523–6.667) | $2.42 \times 10^{-66}$ |
| | Delta: 1.351 (1.11–1.643) | $2.68 \times 10^{-3}$ |
| | Omicron1: 0.837 (0.681–1.029) | $9.06 \times 10^{-2}$ |
| | Omicron2: 0.781 (0.637– 0.957) | $1.73 \times 10^{-2}$ |
| T2D + SARS-CoV-2 variants + Vaccine + Clinical | T2D: 1.879 (1.716–2.057) | $3.20 \times 10^{-42}$ |
| | Early: 5.221 (4.325–6.302) | $2.02 \times 10^{-66}$ |
| | EU1: 2.822 (2.316–3.438) | $6.95 \times 10^{-25}$ |
| | Alpha: 3.481 (2.861–4.235) | $1.18 \times 10^{-35}$ |
| | Delta: 1.199 (0.985–1.460) | $6.98 \times 10^{-2}$ |
| | Omicron1: 0.789 (0.642–0.969) | $2.41 \times 10^{-2}$ |
| | Omicron2: 0.796 (0.649– 0.976) | $2.82 \times 10^{-2}$ |
| | Vaccine: 0.262 (0.226–0.303) | $2.08 \times 10^{-71}$ |

Clinical: age, gender, BMI, genotyping array, PC 1–4.
*HR* hazard ratio, *CI* confidence interval.

CI = 0.689–1.040), and 0.789 (95% CI = 0.644–0.968), respectively, indicating that the fatality rate was highest in the early variants, followed by Alpha, EU1, Delta, and Omicron variants.

Figure 4a and b show the estimated survival curves stratified with T2D and PRS groups based on the Survival-COVID-19 model, respectively. We found that T2D patients and controls had significantly different ($P < 0.05$) survival probabilities and the survival probabilities of the three PRS groups were also significantly different ($P < 0.05$), according to pairwise log-rank tests. To study the difference in survival probability between individuals aged below 60 years and above 60 years, we divided participants into two groups based on their ages. Survival curves stratified by age groups and T2D are displayed in Fig. 4c, and survival curves stratified by age groups and PRS groups for T2D are in Fig. 4d. Figure 4c and d clearly show that survival probabilities for T2D patients, or individuals with a high genetic predisposition for T2D in the aged population, decreased much more than those in the younger population when compared to controls.

**Assessment of survival model performances**. To assess the performance of survival models, we computed the concordance index (C-index), which represents the proportion of observation pairs for which estimated risk scores based on Cox PH models correspond to their survival times. Based on 10-fold CV, we estimated Cox PH model-based risk scores for all individuals and compared their survival times to estimate the C-index (Supplementary Table 5). The C-index for the Survival-All model was 0.770 for T2D and 0.755 for T2D PRS groups, suggesting that, in rough terms, given two individuals from the data, the survival model can predict with 77% or 75.5% accuracy who will die first. The C-index for the Survival-COVID-19 model was 0.791 for T2D and 0.795 for T2D PRS groups.

**Sensitivity analysis**. In the sensitivity analysis, we computed individual T2D PRS based on the GWAS result from the Diabetes Genetics Replication And Meta-analysis (DIAGRAM) consortium for 459,119 participants[30]. To mitigate the confounding effect of BMI on T2D, we utilized T2D summary statistics adjusted for BMI from the DIAGRAM consortium, despite its smaller sample size of 26,676 T2D cases and 132,532 controls, compared to the BMI-unadjusted summary statistics. Supplementary Fig. 8 displays the scatter plot of T2D PRS between UK Biobank-derived PRS and DIAGRAM-derived PRS. We included three PRS groups (low, medium, and high) based on the DIAGRAM-based T2D PRS as covariates in both the Survival-All and the Survival-COVID-19 models. The estimated survival curves, stratified by T2D and PRS groups, as well as survival curves stratified by PRS groups, are represented in Supplementary

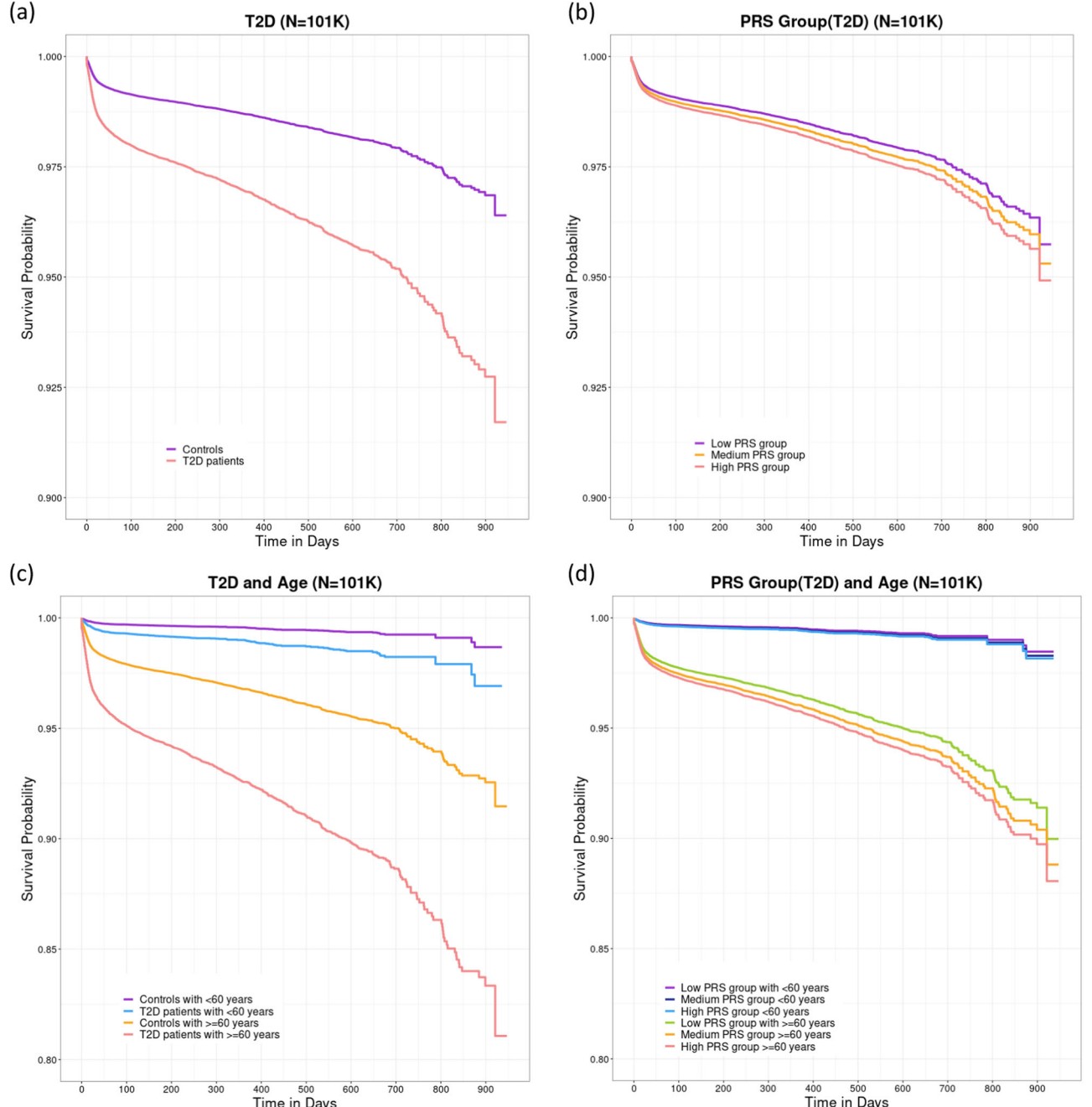

**Fig. 4 The survival curves, stratified by T2D or T2D PRS and SARS-CoV-2 infection, based on the Survival-COVID-19 model, as well as the survival curves stratified by age groups and T2D or T2D PRS. a** The survival curves stratified by T2D ($N = 101K$) based on the Survival-COVID-19 model, adjusted for clinical covariates such as age, BMI, gender, genotyping array, and PC 1–4. **b** The survival curves stratified by PRS groups for T2D ($N = 101K$) based on the Survival-COVID-19 model, adjusted for clinical covariates. **c** The survival curves stratified by age groups and T2D. **d** The survival curves stratified by age groups and PRS groups for T2D.

Fig. 9a and b, respectively. The overall patterns of the survival curves are similar to the corresponding curves in Figs. 3b and 4b.

We presented the results for the association of T2D and T2D PRS using proportional odds models in Table 2, and the effects of T2D and T2D PRS on SARS-CoV-2 infection using logistic models in Supplementary Note 2, Supplementary Tables 6 and 7. The results from these two models differ significantly because the proportional odds models predict the three categories (i.e., moderate, severe, and critical) for COVID-19 severity in individuals infected with SARS-CoV-2, whereas the logistic

models only consider the SARS-CoV-2 infection status among all participants. Given that over 22% of the current UK Biobank samples were infected with SARS-CoV-2, the infection status is no longer significantly associated with T2D and T2D PRS, although it was significantly associated during the initial phase of the COVID-19 pandemic due to its rarity. However, the severity of COVID-19 remains strongly associated with T2D and T2D PRS. Therefore, we primarily focused on the association between T2D and severe COVID-19, rather than on SARS-CoV-2 infection.

**Partitioned T2D PRS**. We constructed individual-level partitioned T2D PRS to capture the five physio-pathological components of T2D, namely beta-cell function, proinsulin, obesity, lipodystrophy, and liver/lipid metabolism[31]. The SNP lists and weights for these cluster-specific partitioned T2D PRS are detailed in Supplementary Table 8. Utilizing these SNP lists and weights, we calculated the partitioned T2D PRS for UK Biobank samples, presenting the AUC values in Supplementary Table 9. Subsequently, we conducted analyses using the newly estimated partitioned T2D PRS with proportional odds models and survival models. In Supplementary Table 10, the P-values for the OR of the beta-cell and proinsulin PRSs in the proportional odds models were not significant. In contrast, the P-values for the obesity, lipodystrophy, and liver/lipid PRSs were significant. This indicates that only PRSs for traits associated with insulin resistance (i.e. obesity, lipodystrophy, and liver/lipid), not insulin production and secretion (i.e. beta-cell and proinsulin), are significantly related to the severity of COVID-19. However, in Supplementary Tables 11 and 12, no P-values were significant, suggesting that none of the partitioned T2D PRSs are significantly associated with the mortality of COVID-19.

## Discussion

In this paper, we investigated the association between T2D or genetic susceptibility to T2D, and COVID-19 severity using proportional odds regression and then systemically examine the impact of T2D in patients who tested positive for COVID-19 on survival time using multivariable Cox PH models with all individuals or only SARS-CoV-2-infected individuals. In the proportional odds models, we found a significant association between the risks of T2D or genetic susceptibility to T2D and the severity of COVID-19. This suggests that individuals with T2D or those genetically predisposed to T2D have a higher likelihood of experiencing severe COVID-19 symptoms, pointing to a role for T2D and T2D-related genetics in the pathogenesis of COVID-19 severity. In the Cox PH model with all individuals, we found that the risk of death for T2D patients and individuals in the high T2D PRS group were higher than for controls. Moreover, SARS-CoV-2-infected individuals with T2D or relatively high genetic risks for T2D showed a significant correlation with increased mortality, indicating a role for genetic interplay in the underlying pathogenesis of T2D and COVID-19. Lastly, when working only with SARS-CoV-2-infected individuals, we studied the association of T2D and T2D PRS with survival times and found that T2D patients who tested positive for COVID-19 had a significantly higher mortality rate than non-diabetic individuals who tested positive for COVID-19. We also discovered that the fatality rate was highest for the early variants, succeeded by the Alpha, EU1, Delta, and Omicron variants and the odds of severe COVID-19 were substantially lower in the vaccinated group compared to the non-vaccinated group.

The use of PRS in identifying the relationship between genetic susceptibility for T2D and severe COVID-19 has several clinical and research implications. First, our results underscored the importance of considering T2D status and genetic susceptibility to T2D in managing COVID-19. Early detection and management of COVID-19 in patients with T2D or high genetic risk for T2D might improve clinical outcomes. Second, in our analysis, we found participants who did not have T2D but were classified in the high PRS group for T2D were more likely to have severe COVID-19. It suggested the potential use of PRS to classify COVID-19 patients for better medical care since the T2D PRS proved useful in estimating the severity and mortality of COVID-19, even amongst individuals who did not currently have T2D. Third, the observed positive relationship between T2D PRS and COVID-19 provides an avenue for discovering the causal impact of novel genetic factors on

the development of severe COVID-19. Lastly, the biological profiling of SARS-CoV-2-infected T2D patients may uncover the molecular mechanism underlying the interaction between host genetics and the pathogenesis of SARS-CoV-2 that links T2D to severe COVID-19. This could ultimately suggest new methods for prevention, prediction, and medical therapy for such patients.

This study has several limitations. First, we analyzed only individuals infected with SARS-CoV-2 prior to December 2022. The number of Omicron cases continued to increase after December 2022, and thus further analysis of these cases is necessary to evaluate the effect of T2D and susceptibility to T2D on the severity and mortality of COVID-19. Second, a variety of vaccines (e.g., Pfizer, Moderna, and Johnson & Johnson) have been continuously utilized worldwide, and thus we need to study the effect of these COVID-19 vaccines on the mortality rate. Thrid, we only included the Caucasian population, with an average age of 58 years and thus we should be cautious about generalizing these findings to other populations or age groups. Fourth, we found high genetic correlation between T2D and severity of COVID-19 and the usefulness of T2D PRS in estimating the severity and mortality of COVID-19 but we did not further investigate the shared genetic architecture between T2D and COVID-19. We are currently identifying their shared genetic loci based on cross-trait meta-analysis[32] and detecting gene expression associations in specific tissues for T2D and COVID-19 based on transcriptome-wise association study (TWAS)[33]. Fifth, despite the high genetic correlation between T2D and severe COVID-19, the heritability of severe COVID-19 is considerably low (~3%) compared to that of T2D (~15%). Although the genetic correlation between T2D and severe COVID-19 is statistically significant, further careful investigation is needed to interpret the shared genetic architecture between them. Lastly, while our analysis primarily focused on the relationship between T2D and severe COVID-19 due to their well-established association, it is necessary to further examine the association between other underlying diseases (i.e. respiratory diseases, hypertension, cardiovascular disease, dementia) and severe COVID-19, given the considerable genetic correlation between these diseases and severe COVID-19.

In conclusion, our study was undertaken to investigate the association between T2D, or genetic susceptibility to T2D, and COVID-19 severity using UK Biobank data. We aimed to understand the impact of T2D in patients who tested positive for COVID-19 on survival time, and how genetic risks for T2D, calculated using individual PRS, influenced these outcomes. We found that T2D and genetic predisposition for T2D were significantly associated with the risk of COVID-19 severity, and an increased mortality rate and T2D PRS proved useful in estimating the severity and mortality of COVID-19. These observations should help clinicians categorize patients based on their underlying diseases (i.e. T2D), their PRS as well as the severity of COVID-19, to provide more effective treatment. Lastly, our findings should aid further research to delineate the complex interactions between SARS-CoV-2 infections, COVID-19 severity, airway inflammation, and other outcomes in patients with T2D.

## Methods

**Study population and design**. The UK Biobank is a large-scale cohort study that includes over 500,000 participants aged between 40 and 69 at the time of recruitment from the UK during the period of 2006–2010[34]. For our analysis, we utilized questionnaire-based self-reported data, hospital inpatient data, and death data, all updated as of December 19, 2022. We also incorporated COVID-19 test result data, which included records of polymerase chain reaction (PCR) tests for SARS-CoV-2. We

restricted our study to only individuals of European ancestry based on the information of self-reported ethnicity, resulting in 459,119 individuals. The SNPs were filtered out if they had a minor allele frequency (MAF) of <0.01 and an imputation quality score of <0.8, leading to 9,572,556 SNPs.

**Heritability and genetic correlations**. The GWAS summary statistics of T2D and related traits were produced from a large-scale cohort data of 459,119 participants in the UK Biobank. These statistics were adjusted for potential confounders such as age, age squared, gender, genotype PCs, assessment array, and genotyping array using BOLT-LMM v2.3 software[35]. To estimate heritability and genetic correlations between T2D, its related traits, and COVID-19, we performed cross-trait LD score regression (LDSC) analysis based on GWAS summary statistics using the LDSC software[36]. We utilized the 1000 Genomes project data (phase 3 version 5) as a reference panel and restricted our study to those of European ancestry with common and well-imputed HapMap3 SNPs. We estimated heritability and genetic correlation between various traits, including T2D, SARS-CoV-2 infection, and severe COVID-19.

**Polygenic risk score (PRS)**. The PRS represents an individual's genetic liability for a particular trait, calculated by multiplying the genotype dosage for each variant by its corresponding weight and summing across all variants[37,38]. To compute the individual PRS of 459,119 participants, we utilized a 10-fold cross-validation (CV) approach. We first generated 10 GWAS summary statistics with 10 training sets for each trait, then computed PRS for 10 validation sets using only HapMap3 SNPs using the LDpred method[39], thus obtaining PRS scores for all individuals. We merged the PRS scores for the 10 folds after standardizing the PRS in each fold to avoid bias. The candidate PRSs were calculated with a range of tuning parameters (i.e., the fraction of causal variants), and the PRS with the b est discriminative capability was selected based on the maximal area under the curve (AUC).

**Proportional odds regression**. To investigate the effects of T2D and genetic susceptibility for T2D on the severity of COVID-19, we divided participants into three groups according to the severity of their COVID-19 symptoms: mild (1), moderate (2), and critical (3). If samples were obtained from the participants were hospitalized with COVID-19, they were classified as having moderate COVID-19. If the participants died due to COVID-19, they were classified as having critical COVID-19. Otherwise, they were classified as having mild COVID-19. With the proportional odds assumption, we performed a proportional odds regression for T2D or T2D PRS, and the analysis proceeded separately for T2D or T2D PRS: $logit(P(Y_i \leq j)) = \beta_{0j} + \beta_1 X_i + \boldsymbol{\beta_2 C_i}$ for $j = 1, 2, 3$ where $Y_i$ is an ordinal outcome for COVID-19 severity (1:mild, 2: moderate, 3:critical) of $i$th individual; $j$ is the severity of COVID-19; $X_i$ is a T2D or T2D PRS; $\boldsymbol{C_i}$ is a set of clinical covariates including age, gender, vaccination status, BMI, genotyping array, and the top four PCs. We also included an indicator for the three PRS groups for T2D (i.e., low, medium, and high) as a covariate (with low as the reference) in the model. The assessment of model performances and the assumption of constant effect sizes of risk factors were tested using Brant tests.

**Survival analysis with all participants**. To examine the effect of T2D and SARS-CoV-2 infection on survival time, we performed survival analysis using multivariable Cox proportional hazards (PH) models with all participants (Survival-All). We excluded individuals who died before and after the study period, resulting in a total of 431,975 participants included in the study. Out of

these, 12,911 participants (2.99%) died during the study period, and the remaining ones (97.01%) were censored. Survival time was defined as the number of days from the start date of the study to the date of death or the last date of the study. The effect of SARS-CoV-2 infection is not constant over the study period, which is a violation of the proportional hazards assumption. Thus, we divided the analysis time into five time-intervals based on SARS-CoV-2 variants such as Delta (494 days until the proportion of the Delta variant had exceeded 95%), Omicron1 (672), Omicron2 (757), and Omicron3 (851) and Cox proportional hazards model was stratified for these time-intervals[28]. The effect of SARS-CoV-2 infection varied over time, which can be explored via stratification by time. We fit the following multivariable Cox proportional hazards models with time-dependent coefficients: $h(t|\boldsymbol{x_i}) = h_0(t)\exp(\beta_1 X_{1i} + \boldsymbol{\beta_2 X_{2i}} + \boldsymbol{\beta_3 C_i})$ where $X_{1i}$ is a T2D or T2D PRS; $\boldsymbol{X_{2i}}$ is a set of indicators for one of the five time-intervals based on SARS-CoV-2 variants (i.e. $\boldsymbol{X_{2i}} = (I(t \leq 494), I(494 < t \leq 672), I(672 < t \leq 757), I(757 < t \leq 851), I(851 < t))^{\mathrm{T}})$; $\boldsymbol{C_i}$ is a set of clinical covariates. We checked the proportional hazards assumption of this stratified Cox regression model. Again, we included an indicator for the three PRS groups for T2D (i.e., low, medium, and high) as a covariate in the model. We performed stratified Cox PH analysis and pairwise log-rank tests to assess the differences in survival probabilities for groups stratified by T2D and SARS-CoV-2 infection.

**Survival analysis with only SARS-CoV-2-infected individuals**. We further performed survival analysis with only SARS-CoV-2-infected individuals (Survival-COVID-19). For this analysis, we only included individuals with confirmed SARS-CoV-2 infection dates, leading to a total of 101,086 patients included in the study. Among these participants, the number of deaths was 2,836 (2.8%), while the number of censored individuals was 98,250 (97.2%). For the Survival-COVID-19 model, survival time was defined as the number of days from the date of SARS-CoV-2 infection to the date of death or the last date of the study. We performed multivariable Cox PH models, similar to the Survival-All model, but excluded the indicator for SARS-CoV-2 infection, as we only considered SARS-CoV-2-infected individuals for the Survival-COVID-19 model. Instead, we included an indicator for SARS-CoV-2 variants as a covariate in the model to investigate the effects of SARS-CoV-2 variants on the mortality rate. We also estimated survival probability stratified by T2D or T2D PRS groups, and the differences in survival probabilities across the groups were assessed using pairwise log-rank tests.

**Statistics and reproducibility**. Statistical analyses and figures were created using R version 4.2.3. We conducted the statistical analyses using multivariable logistic regression models, proportional odds models, and Cox proportional hazards models. All the codes used are publicly available on GitHub (https://github.com/wonilchung/CovidT2D) to guarantee the reproducibility of all analyses.

**Reporting summary**. Further information on research design is available in the Nature Portfolio Reporting Summary linked to this article.

### Data availability

All data are available for academic uses at https://github.com/wonilchung/CovidT2D. All other data are available from the corresponding author upon reasonable request.

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

## Acknowledgements

We appreciate the individuals who participated in the study. This research has been conducted using the UK Biobank Resource (application numbers 45052, 58105, 77890). This research was supported by the Bio and Medical Technology Development Program of the National Research Foundation (NRF) funded by the Korean government (2021M3E5E3081425). This work was also supported by the NRF grant funded by the Korea government (2020R1C1C1A01012657) and Basic Science Research Program of the NRF funded by the Korea government (2021R1A6A1A10044154). This work was also supported by the Soongsil University Research Fund.

## Author contributions

A.L., L.L., T.P., and W.C. conceived and designed the experiments. A.L., J.S., S.P., Y.C., G.K., J.L., and W.C. performed the experiments and analyzed the data. They were also responsible for generating the necessary data, materials, and analysis tools. A.L. and W.C. primarily wrote the manuscript. A.L., L.L., T.P., and W.C. reviewed and revised the manuscript.

## Competing interests

The authors declare no competing interests.
