## [Peer Review File · Communications Biology]

Reviewers' comments:

Reviewer #1 (Remarks to the Author):

This manuscript aimed at evaluating the impact of type 2 diabetes (T2D) and its genetic susceptibility on the severity and mortality of COVID-19 using data from 500K individuals in the UK Biobank. The authors identified a significant association between T2D polygenic risk score, and COVID-19 severity, and suggested a pivotal role of T2D-related genetics in the pathogenesis of severe COVID-19. They also found that individuals in the high T2D PRS group had a significantly increased mortality rate compared to others.

Although this manuscript is overall well-written and the study design well-conducted, I have a couple of major suggestions to submit.

First, there have been previous Mendelian randomization studies looking at the effects of genetically determined T2D and severity of covid infection; results have been inconsistent with respect to the direct causal effect; some of them pointed out the confounding effect of BMI. In this study, body mass index (BMI) has not been used as a covariate in the GWAS performed to build the T2D PRS in the primary analyses. Moreover, in the sensitivity analyses, when replicating the analyses using DIAGRAM summary statistics as inputs to calculate the T2D PRS, estimates of T2D used for each SNP were not adjusted for BMI (while BMI-adjusted estimates are publicly available).

Second, while the usage of T2D PRS has been well-established to assess the overall T2D genetic susceptibility, a partitioned T2D PRS capturing the various physiopathological components of T2D separately (beta-cell failure, insulin resistance, etc, .., as described by DiCorpo et al, Diab Care, 2022) would have provided more precise evidence on the share biological pathways between T2D and COVID severity.

Detailed suggestions:

-The introduction should summarize previous findings from Mendelian Randomization studies on the association between T2D and covid severity, and better position the novelty of this current work

-Methods:

--Primary and sensitivity analyses should account for BMI in the calculation of the T2D PRS to avoid a confounding effect of BMI on the reported association between T2D PRS and covid severity; BMI should not be only used as a covariate in the clinical models, but in the PRS calculation given the important genetic overlap between T2D and BMI

--I suggest performing additional analyses using partitioned T2D PRS to investigate the share biological pathways between T2D and covid severity

-Results/discussion: the estimated heritability of covid severity is extremely low (~3%); the authors should discuss the limitations due to this low heritability when running the genetic correlations with T2D

Reviewer #2 (Remarks to the Author):

The study evaluated the impact of type 2 diabetes (T2D) and its genetic susceptibility on the severity and mortality of COVID-19 using data from 459,119 individuals in the UK Biobank. There are two questions.

1. Supplementary Table 1 showed the heritability and genetic correlations between various traits including T2D, SARS-CoV-2 infection and severe COVID-19. We have found several other strongly correlated PRSs associated with SARS-CoV-2. Why did you choose T2D? Please explain in the main text.

2. Table 2 showed the results of proportional odds models for the association of severity of COVID-19 and T2D. You used the proportional odds models. In the Supplementary Table 6, you showed the results of multivariable logistic regression models representing the effects of T2D and T2D PRS on SARS-CoV-2 infection. You used the multivariable logistic regression models. And the results were heterogeneous. Can you elaborate on their differences? Why use different models? What causes this difference?

Response to Reviewers,

We are grateful to the reviewers for their positive feedback and thorough critique of our manuscript. Their insightful comments enabled us to perform a substantial revision, significantly improving the manuscript's quality. In this revision, we first re-conducted the GWAS analyses for T2D, incorporating BMI as an additional covariate. We then re-performed all analyses with re-calculated T2D PRS to mitigate the confounding effect of BMI on the association between T2D PRS and COVID-19 severity. Furthermore, we computed partitioned T2D PRS for five physio-pathological components of T2D—beta-cell function, proinsulin levels, obesity, lipodystrophy, and liver/lipid metabolism—and investigated their association with severity and mortality of COVID-19. This revision addresses all comments from the reviewers and fulfills the editorial requests we received. For your convenience, we have used track changes in Microsoft Word to highlight all textual updates in the draft and provide a detailed, point-by-point reply to each reviewer comment, with our responses highlighted in blue below.

Reviewer #1 (Remarks to the Author):

This manuscript aimed at evaluating the impact of type 2 diabetes (T2D) and its genetic susceptibility on the severity and mortality of COVID-19 using data from 500K individuals in the UK Biobank. The authors identified a significant association between T2D polygenic risk score, and COVID-19 severity, and suggested a pivotal role of T2D-related genetics in the pathogenesis of severe COVID-19. They also found that individuals in the high T2D PRS group had a significantly increased mortality rate compared to others.

Although this manuscript is overall well-written and the study design well-conducted, I have a couple of major suggestions to submit.

We would like to thank the reviewer for their positive remarks and for concisely summarizing our work. Following the reviewer's suggestions, we have conducted the following new analyses.

First, there have been previous Mendelian randomization studies looking at the effects of genetically determined T2D and severity of covid infection; results have been inconsistent with respect to the direct causal effect; some of them pointed out the confounding effect of BMI. In this study, body mass index (BMI) has not been used as a covariate in the GWAS performed to build the T2D PRS in the primary analyses.

As the reviewer pointed out, previous Mendelian Randomization (MR) studies on the causal effects of T2D on COVID-19 severity have shown inconsistent results but they all acknowledged the confounding effect of BMI on T2D. Consequently, following the reviewer's suggestion, we included BMI as an additional covariate in the GWAS to construct the T2D PRS. For our primary analysis, we conducted GWAS for T2D, adjusted additionally for BMI, using 459K UK Biobank samples. The results are presented below. The left panel depicts GWAS results unadjusted for BMI, while the right panel shows results adjusted for BMI. The overall patterns are similar, but generally, the effects are more significant in the BMI-adjusted results. Additionally, we noted that several significant peaks—specifically at Chr2:600575-653874, Chr3:49774658-50160109, and Chr16:53797908-70006767—were not present in the new results. We discovered that these regions are associated with obesity, BMI, waist-hip ratio, and HDL cholesterol, as identified in the GWAS catalog.

The scatter plots below compare the GWAS results unadjusted for BMI with those adjusted for BMI. Generally, the results adjusted for BMI demonstrate more significance. However, several SNPs on chromosomes 2, 3, and 16 are significant only in the unadjusted results, as mentioned previously. The right panel of the plots provided below shows these details more explicitly.

For our sensitivity analyses, we initially utilized DIAGRAM summary statistics titled “Summary of T2D associations, unadjusted for BMI and without UK Biobank subjects” from Mahajan et al. (2018). However, to incorporate T2D summary statistics adjusted for BMI, we switched to using DIAGRAM summary statistics titled “DIAGRAM 1000G GWAS meta-analysis with adjustment for BMI Stage 1 Summary Statistics” from Scott et al. (2017). The BMI-adjusted summary statistics have a relatively smaller sample size, comprising 26,676 T2D cases and 132,532 controls, compared to the 74,124 T2D cases and 824,006 controls in the BMI-unadjusted summary statistics. Despite the smaller sample size, we opted for the BMI-adjusted summary statistics for our sensitivity analysis, following the reviewer's suggestion. Similar to the UK Biobank summary statistics, the results from both datasets appear similar (as shown below), but those adjusted for BMI are generally less significant, primarily due to the smaller sample size.

The scatter plots below compare the DIAGRAM summary statistics unadjusted for BMI with those adjusted for BMI. Generally, the BMI-adjusted DIAGRAM summary statistics demonstrate less significance. Additionally, several SNPs on chromosomes 2, 3, and 16 are significant only in those unadjusted for BMI, a pattern that is similar to the UKB summary statistics.

Second, while the usage of T2D PRS has been well-established to assess the overall T2D genetic susceptibility, a partitioned T2D PRS capturing the various physiopathological components of T2D separately (beta-cell failure, insulin resistance, etc., ..., as described by DiCorpo et al, Diab Care, 2022) would have provided more precise evidence on the share biological pathways between T2D and COVID severity.

We greatly appreciate the reviewer's suggestion. We constructed individual-level partitioned T2D PRS to capture the five physio-pathological components of T2D, namely beta-cell function, proinsulin, obesity, lipodystrophy, and liver/lipid metabolism, based on DiCorpo et al. (2022). The SNP lists and weights for these cluster-specific partitioned T2D PRS are detailed in Supplementary Table 8. Using these SNP lists and weights with Plink software, we calculated the partitioned T2D PRS for UK Biobank samples, presenting AUC values in Supplementary Table 9. The AUCs for the partitioned T2D PRS, adjusted for clinical covariates (age, gender, BMI, genotyping array, and the top 4 PCs), are 0.671, 0.661, 0.660, 0.664, and 0.660 for beta-cell, proinsulin, obesity, lipodystrophy, and liver/lipid components, respectively. Notably, the beta-cell and proinsulin components exhibit larger AUCs compared to the other three components (obesity, lipodystrophy, and liver/lipid). This discrepancy may be attributed to the larger number of SNPs used to build the partitioned PRS for the beta-cell and proinsulin clusters. Moreover, as reported in DiCorpo et al. (2022), these two components are associated with insulin production and secretion, while the other three are linked to insulin resistance.

Detailed suggestions:

-The introduction should summarize previous findings from Mendelian Randomization studies on the association between T2D and covid severity, and better position the novelty of this current work

We thank the reviewer for the detailed suggestions. In response, we have added the following sentences to the Introduction and Results sections of the manuscript:

In the Introduction, we added, “First, we estimate the genetic predisposition for T2D by computing the T2D PRS using GWAS summary statistics from the UK Biobank. We include body mass index (BMI) as a covariate in the GWAS when constructing the T2D PRS, to mitigate the confounding effect of BMI on T2D. This approach is informed by the Mendelian Randomization (MR) studies on the causal effects of T2D on COVID-19 severity, which have yielded inconsistent results, largely attributable to the confounding impact of BMI on T2D.”

In the Results section, we added, “In the sensitivity analysis, we computed individual T2D PRS based on the GWAS result from the Diabetes Genetics Replication And Meta-analysis (DIAGRAM) consortium for 459,119

participants. To mitigate the confounding effect of BMI on T2D, we utilized T2D summary statistics adjusted for BMI from the DIAGRAM consortium, despite its smaller sample size of 26,676 T2D cases and 132,532 controls, compared to the BMI-unadjusted summary statistics.”

-Methods:

--Primary and sensitivity analyses should account for BMI in the calculation of the T2D PRS to avoid a confounding effect of BMI on the reported association between T2D PRS and covid severity; BMI should not be only used as a covariate in the clinical models, but in the PRS calculation given the important genetic overlap between T2D and BMI

Following the reviewer's suggestion, we generated new T2D PRS scores using LDpred, based on the GWAS summary statistics adjusted for BMI. We have fully reconducted the analyses with the newly computed PRS scores using proportional odds models and survival models. The updated results are summarized below, and Tables 2, 3, and 4 in the manuscript have been modified accordingly. As indicated below, the Odds Ratios (OR) for the new T2D PRS are smaller than those for the previous T2D PRS. Similarly, the p-values for the new T2D PRS are generally less significant than those for the previous T2D PRS. This change can be attributed to the removal of BMI's confounding effect from the T2D GWAS.

Tables	Covariates	OR / HR (Prev)	OR / HR (New)	P-value (Prev)	P-value (New)
Table 2	T2D PRS + Clinical	T2D PRS: 1.105	T2D PRS: 1.080	4.26E-13	8.58E-08
	T2D PRS Group + Clinical	Medium: 1.095	Medium: 1.083	4.00E-03	1.66E-02
		High: 1.301	High: 1.211	7.90E-10	7.01E-05
Table 3	T2D PRS + Clinical	T2D PRS: 1.080	T2D PRS: 1.067	7.38E-18	4.95E-12
	T2D PRS + SARS-CoV-2 + Clinical	T2D PRS: 1.073	T2D PRS: 1.062	4.95E-15	1.31E-10
	T2D PRS Group + Clinical	Medium: 1.035	Medium: 1.077	3.71E-06	3.19E-03
		High: 1.245	High: 1.192	6.59E-14	1.53E-07
	T2D PRS Group + SARS-CoV-2 + Clinical	Medium: 1.106	Medium: 1.075	7.20E-06	3.74E-03
High: 1.224	High: 1.181	7.43E-12	7.37E-07		
Table 4	T2D PRS + Clinical	T2D PRS: 1.083	T2D PRS: 1.062	1.96E-05	1.99E-03
	T2D PRS Group + Clinical	Medium: 1.058	Medium: 1.089	2.53E-01	1.23E-01
		High: 1.242	High: 1.190	4.16E-04	1.55E-02
	T2D PRS + SARS-CoV-2 variants + Clinical	T2D PRS: 1.054	T2D PRS: 1.045	4.95E-03	2.31E-02
	T2D PRS Group + SARS-CoV-2 variants + Clinical	Medium: 1.059	Medium: 1.093	2.41E-01	1.09E-01
High: 1.164		High: 1.145	1.36E-02	6.12E-02	

--I suggest performing additional analyses using partitioned T2D PRS to investigate the share biological pathways between T2D and covid severity

As suggested by the reviewer, we conducted additional analyses using newly estimated partitioned T2D PRS with proportional odds models and survival models. We summarized these results below and included them in Supplementary Tables 10, 11, and 12. In Supplementary Table 10, the p-values for OR of the beta-cell and proinsulin PRSs in the proportional odds models were not significant, whereas those for obesity, lipodystrophy, and liver/lipid PRSs were significant. This suggests that only PRSs for traits associated with insulin resistance, namely obesity, lipodystrophy, and liver/lipid, are significantly related to the severity of COVID-19. However,

in Supplementary Tables 11 and 12, no p-values are significant, indicating that none of the partitioned T2D PRSs are significantly associated with the mortality of COVID-19. We have added some sentences to the Results section of the manuscript to reflect these findings.

In the Results section, we added, “We constructed individual-level partitioned T2D PRS to capture the five physio-pathological components of T2D, namely beta-cell function, proinsulin, obesity, lipodystrophy, and liver/lipid metabolism. The SNP lists and weights for these cluster-specific partitioned T2D PRS are detailed in Supplementary Table 8. Utilizing these SNP lists and weights, we calculated the partitioned T2D PRS for UK Biobank samples, presenting the AUC values in Supplementary Table 9. Subsequently, we conducted analyses using the newly estimated partitioned T2D PRS with proportional odds models and survival models. In Supplementary Table 10, the p-values for the OR of the beta-cell and proinsulin PRSs in the proportional odds models were not significant. In contrast, the p-values for the obesity, lipodystrophy, and liver/lipid PRSs were significant. This indicates that only PRSs for traits associated with insulin resistance (i.e. obesity, lipodystrophy, and liver/lipid), not insulin production and secretion (i.e. beta-cell and proinsulin), are significantly related to the severity of COVID-19. However, in Supplementary Tables 11 and 12, no p-values were significant, suggesting that none of the partitioned T2D PRSs are significantly associated with the mortality of COVID-19.”

Supp Tables	Cluster	Covariates	OR / HR (95% CI)	P-value
Supp Table 10	BetaCell	PRS + Clinical	PRS: 1.032 (0.986 - 1.08)	1.74E-01
	Proinsulin	PRS + Clinical	PRS: 1.07 (0.991 - 1.156)	8.50E-02
	Obesity	PRS + Clinical	PRS: 1.14 (1.05 - 1.237)	1.70E-03
	Lipodystrophy	PRS + Clinical	PRS: 1.09 (1.025 - 1.158)	5.56E-03
	Liver/Lipid	PRS + Clinical	PRS: 1.166 (1.065 - 1.276)	8.65E-04
Supp Table 11	BetaCell	PRS + Clinical	PRS: 1.013 (0.982 - 1.045)	4.01E-01
	Proinsulin	PRS + Clinical	PRS: 0.973 (0.917 - 1.033)	3.78E-01
	Obesity	PRS + Clinical	PRS: 1.026 (0.961 - 1.095)	4.50E-01
	Lipodystrophy	PRS + Clinical	PRS: 1.033 (0.989 - 1.079)	1.44E-01
	Liver/Lipid	PRS + Clinical	PRS: 0.985 (0.913- 1.064)	7.04E-01
Supp Table 12	BetaCell	PRS + Clinical	PRS: 1.033 (0.969 - 1.101)	3.22E-01
	Proinsulin	PRS + Clinical	PRS: 0.934 (0.825 - 1.057)	2.79E-01
	Obesity	PRS + Clinical	PRS: 1.02 (0.891 - 1.167)	7.77E-01
	Lipodystrophy	PRS + Clinical	PRS: 1.013 (0.925 - 1.109)	7.87E-01
	Liver/Lipid	PRS + Clinical	PRS: 0.859 (0.73 - 1.011)	6.60E-02

-Results/discussion: the estimated heritability of covid severity is extremely low (~3%); the authors should discuss the limitations due to this low heritability when running the genetic correlations with T2D

We greatly appreciate the reviewer's suggestion. We addressed the issue of low heritability of COVID-19 in the discussion section as follows:

“Fifth, despite the high genetic correlation between T2D and severe COVID-19, the heritability of severe COVID-19 is considerably low (approximately 3%) compared to that of T2D (approximately 15%). Although the genetic correlation between T2D and severe COVID-19 is statistically significant, further careful investigation is needed to interpret the shared genetic architecture between them.”

Response to Reviewers,

We are grateful to the reviewers for their feedback and thorough critique of our manuscript. Their insightful comments enabled us to perform a substantial revision, significantly improving the manuscript's quality. In this revision, we have included an explanation for our focus on T2D and T2D PRS, rather than other underlying diseases, and the rationale behind studying the association between T2D and COVID-19 severity, rather than SARS-CoV-2 infection. This revision addresses all comments from the reviewers and fulfills the editorial requests we received. For your convenience, we have used track changes in Microsoft Word to highlight all textual updates in the draft and provide a detailed, point-by-point reply to each reviewer comment, with our responses highlighted in blue below.

Reviewer #2 (Remarks to the Author):

The study evaluated the impact of type 2 diabetes (T2D) and its genetic susceptibility on the severity and mortality of COVID-19 using data from 459,119 individuals in the UK Biobank. There are two questions. We thank the reviewer for summarizing our work. We provide a point-by-point reply to the reviewer's comments below.

1. Supplementary Table 1 showed the heritability and genetic correlations between various traits including T2D, SARS-CoV-2 infection and severe COVID-19. We have found several other strongly correlated PRSs associated with SARS-CoV-2. Why did you choose T2D? Please explain in the main text.

We greatly appreciate the reviewer's comment. In Supplementary Table 1, we presented the estimated heritability and genetic correlation between various traits, including T2D, SARS-CoV-2 infection, and severe COVID-19. As the reviewer pointed out, there is a high genetic correlation between severe COVID-19 and other underlying diseases, such as Coronary Heart Disease (CHD, 0.7352), Cardiovascular Disease (CVD, 0.8138), and stroke (0.8768). Due to the well-known strong association between T2D, its related traits (e.g., Body Mass Index [BMI], Waist-Hip Ratio [WHR]), and severe COVID-19, we initially chose to study these associations in this manuscript. We are currently investigating the association of severe COVID-19 with other underlying diseases and their PRS, and we plan to summarize these findings in our next manuscript. We have added the following sentences to the discussion section:

“Lastly, while our analysis primarily focused on the relationship between T2D and severe COVID-19 due to their well-established association, it is necessary to further examine the association between other underlying diseases (i.e. respiratory diseases, hypertension, cardiovascular disease, dementia) and severe COVID-19, given the considerable genetic correlation between these diseases and severe COVID-19.”

2. Table 2. showed the results of proportional odds models for the association of severity of COVID-19 and T2D. You used the proportional odds models. In the Supplementary Table 6, you showed the results of multivariable logistic regression models representing the effects of T2D and T2D PRS on SARS-CoV-2 infection. You used the multivariable logistic regression models. And the results were heterogeneous. Can you elaborate on their differences? Why use different models? What causes this difference?

We really thank the reviewer for their insightful comment. As the reviewer pointed out, we presented the results for the association of T2D and T2D PRS using proportional odds models in Table 2, and the effects of T2D and T2D PRS on SARS-CoV-2 infection using logistic models in Supplementary Table 6. The results in these two tables differed significantly because the proportional odds models predicted the three categories (i.e. moderate, severe, and critical) for COVID-19 severity in SARS-CoV-2 infected individuals, whereas the logistic models only considered the SARS-CoV-2 infection status among all participants. Given that over 22% of the current UK Biobank samples were infected with SARS-CoV-2, the infection status is no longer significantly associated with T2D and T2D PRS, although it was significantly associated during the initial phase of the COVID-19

pandemic due to its rarity. However, the severity of COVID-19 remains strongly associated with T2D and T2D PRS. Therefore, in the manuscript, we primarily focused on the association between T2D and severe COVID-19, rather than on SARS-CoV-2 infection. We have added the following sentences to the supplementary document:

“We presented the results for the association of T2D and T2D PRS using proportional odds models in Table 2, and the effects of T2D and T2D PRS on SARS-CoV-2 infection using logistic models in Supplementary Table 6. The results in these two tables differ significantly because the proportional odds models predict the three categories (i.e., moderate, severe, and critical) for COVID-19 severity in individuals infected with SARS-CoV-2, whereas the logistic models only consider the SARS-CoV-2 infection status among all participants. Given that over 22% of the current UK Biobank samples were infected with SARS-CoV-2, the infection status is no longer significantly associated with T2D and T2D PRS, although it was significantly associated during the initial phase of the COVID-19 pandemic due to its rarity. However, the severity of COVID-19 remains strongly associated with T2D and T2D PRS. Therefore, in the main manuscript, we primarily focused on the association between T2D and severe COVID-19, rather than on SARS-CoV-2 infection.”

REVIEWERS' COMMENTS:

Reviewer #1 (Remarks to the Author):

The revised version of this manuscript is notably improved and the authors have addressed my comments appropriately; I do not have further suggestions.

Reviewer #2 (Remarks to the Author):

The issue with the article has been resolved